# Biostimulant Properties of Protein Hydrolysates: Recent Advances and Future Challenges

**DOI:** 10.3390/ijms24119714

**Published:** 2023-06-03

**Authors:** Marthe Malécange, Renaud Sergheraert, Béatrice Teulat, Emmanuelle Mounier, Jérémy Lothier, Soulaiman Sakr

**Affiliations:** 1Institut Agro, Univ Angers, INRAE, IRHS, SFR QuaSaV, 49000 Angers, France; marthe.malecange@etud.univ-angers.fr (M.M.); beatrice.teulat@agrocampus-ouest.fr (B.T.); jeremy.lothier@univ-angers.fr (J.L.); 2BCF Life Sciences, Boisel, 56140 Pleucadeuc, France; rsergheraert@bcf-lifesciences.com (R.S.); emounier@bcf-lifesciences.com (E.M.)

**Keywords:** biostimulant, protein hydrolysates, amino acids

## Abstract

Over the past decade, plant biostimulants have been increasingly used in agriculture as environment-friendly tools that improve the sustainability and resilience of crop production systems under environmental stresses. Protein hydrolysates (PHs) are a main category of biostimulants produced by chemical or enzymatic hydrolysis of proteins from animal or plant sources. Mostly composed of amino acids and peptides, PHs have a beneficial effect on multiple physiological processes, including photosynthetic activity, nutrient assimilation and translocation, and also quality parameters. They also seem to have hormone-like activities. Moreover, PHs enhance tolerance to abiotic stresses, notably through the stimulation of protective processes such as cell antioxidant activity and osmotic adjustment. Knowledge on their mode of action, however, is still piecemeal. The aims of this review are as follows: (i) Giving a comprehensive overview of current findings about the hypothetical mechanisms of action of PHs; (ii) Emphasizing the knowledge gaps that deserve to be urgently addressed with a view to efficiently improve the benefits of biostimulants for different plant crops in the context of climate change.

## 1. Introduction

Biostimulants as enhancers of plant vigor and resilience under adverse environmental conditions, are now relevant paths for sustainable agricultural challenges [1]. Agriculture is one of the most sensitive sectors to water scarcity and drought due to global warming [2,3], and it is also compromised by pollution caused by the massive use of mineral fertilizers [4,5]. Within this challenging context, the integration of biostimulants in agriculture is proving to be effective and presently attracts significant interest from the agricultural sector and academic research [6,7,8,9]. A biostimulant is “any substance or microorganism, being able, when applied to plants, to improve nutrition efficiency, abiotic stress tolerance, and/or crop quality traits, regardless of its nutrients content” [10]. Humic and fulvic acids, protein hydrolysates and other N (nitrogen)-containing compounds, seaweeds extracts, chitosan and other biopolymers, inorganic compounds, and beneficial fungi and bacteria are the main types of biostimulants [10,11]. The effects of biostimulants on the mitigation of abiotic stressors [12,13], on plant nutrition [14], as well as on crop quality [15] have been broadly observed and documented in the last decade [16,17,18]. Because of their variable composition, and despite the increasing number of studies conducted worldwide, the mode of action of biostimulants seems to be complex, and drawing up a comprehensive picture of it is still highly challenging. The development of new phenotyping and screening methods and the use of multi-omics approaches will be of great help in piecing together the regulatory network puzzle related to the application of biostimulants [19,20,21,22]. Among all biostimulants, the group of protein hydrolysates (PHs) is of particular interest, as it has positive effects on crop productivity [23]. PHs have been reported to improve plant resilience, particularly by stimulating antioxidant activity within the plant, under adverse environmental conditions [23,24]. Moreover, PHs could act directly on the plant via an adjustment of the carbon and nitrogen metabolisms and the plant hormonal profile, or indirectly via the microbiome [25]. Indeed PHs, provided at the root level (e.g., root dip or drip irrigation) or leaf level (foliar spray), change the microflora community living in the rhizosphere or the phyllosphere [25,26,27,28]. Aside from improving plant accessibility to mineral nutrients, phyllosphere and rhizosphere microorganisms could release enzymes that can convert peptides into smaller fragments acting as signaling compounds to stimulate plant growth [23]. Further in-depth investigations are required to unravel how PHs affect different microbial populations.

The objective of this review is to present the recent advances on PHs’ effects on plant root and shoot development and on the potential mechanisms involved, without considering the origin of PHs’ action (direct or indirect via the microbiome). The relevance of PHs for plants grown under adverse growth conditions is highlighted, and the main physiological changes are discussed. This review also highlights the main knowledge gaps and new research directions that deserve to be explored to optimize their use and efficiency in the future.

## 2. Protein Hydrolysates’ (PHs’) Nature

### 2.1. Origin and Hydrolysis Methods of Raw Materials

PHs correspond to mixtures of amino acids, oligopeptides, and polypeptides, resulting from the partial hydrolysis of different protein sources [29]. Amino acids (AAs) are prevailing components in the formulation of PHs. They are present in a simple form (free AAs) or a complex one (peptides) [17]. The main raw materials used for PH production are effluents and by-products from livestock and the food industry [30,31,32,33]. Their conversion into biostimulants is part of a circular economy approach that contributes to environmental preservation and sustainable agriculture.

PHs are processed by chemical or enzymatic hydrolysis of protein from animal (leather by-products, blood meal, fish by-products, chicken feathers, and casein) or plant (legume seeds, alfalfa hay, and vegetable by-products) sources [23,26]. Most current PH-based biostimulants are produced by chemical hydrolysis of animal-derived proteins and by enzymatic hydrolysis of plant-derived proteins. PHs have distinct chemical characteristics depending on the raw material origin and the production process [34].

### 2.2. Profile of PH Compounds: Free AAs and/or Peptides

The production process and the degree of hydrolysis (DH) determine the basic AA and peptide composition of each biostimulant. PHs produced by enzymatic hydrolysis contain a low proportion of free AAs compared to PHs generated by chemical hydrolysis, but they include more peptides and a high range of AAs. For instance, tryptophan is usually destroyed by chemical (acid) hydrolysis [23]. The enzymatic proteolysis strategy remains more advantageous as it is energy efficient and the final composition of the hydrolysate in AAs and peptides is shaped by the activity and specificity of proteases [33]. Therefore, enzymatically produced plant-derived PHs are mainly characterized by signaling peptides as bioactive compounds, whereas free AAs are the main compounds of chemically produced animal-derived PHs [35] (Figure 1).

### 2.3. Amino Acid Composition of PHs

The main AAs found in PHs depend on the protein source. Glutamic acid seems to be a main component of chicken feather-, fish meal-, casein-, and soybean meal-derived PHs. Aspartic acid dominates in blood meal- and alfalfa hay-derived PHs. Bovine collagen-based PHs have a high content in glycine and proline, and are composed of hydroxyproline and hydroxylysine (two non-standard amino acids), present at negligible levels in plant-derived PHs [23,34]. An analysis of different samples from animal- and plant-derived PHs revealed that histidine and ornithine were absent in the majority of plant-derived PHs studied [34]. Phytotoxicity effects are possible, and even sometimes growth suppression related to the treatment of horticultural crops with PHs. This phenomenon is called ‘general AA inhibition’ and is caused by excessive uptake of free AAs resulting in intracellular AA imbalance, inhibition of nitrate uptake, and increased cell susceptibility to apoptosis [25,35,36,37]. This possible PH inhibitory effect depends not only on the amino acids amount but also on the amino acid supplied and/or on the nutrient medium [37,38]. Aside from AAs and peptides, plant-derived PHs could contain traces of other compounds such as phytohormones, carbohydrates, phenols, and mineral elements [23,39,40].

## 3. Effects of PHs on Plant Development

Extensive studies have reported that the supply of isolated or combined AAs and PHs is beneficial to the vegetative phase of different species such as tomato, beet, or lettuce [41,42,43,44,45] (Figure 1). This effect is also found under constrained growth conditions [46,47,48,49].

### 3.1. Root Architecture

The root architecture—in particular root length and total root area—is paramount to improve water and mineral nutrient use efficiency and in turn plant productivity and resistance to harmful conditions [50,51]. PHs can promote the root development and biomass of various crops (lily, tomato, maize, and potted snapdragon) [52,53,54,55,56,57,58]. One prevailing hypothetical mode of action of PHs implies an auxin-like activity [59]. Auxin is one of the major hormones driving root growth and development [60]. Transcriptomic (mRNAseq) and proteomic analyses revealed that this positive effect may result from the convergence of several physiological changes related to phytohormones, reactive oxygen species (ROS) scavenging, specific primary and secondary metabolic pathways, transport, and cytoskeletal reorganization [51,61]. In order to evaluate the auxin-like activity, the effects of five plant-derived PHs (Solanaceae-, Malvaceae-, Brassicaceae-, and Fabaceae-derived PHs, and a commercial product resulting from the enzymatic hydrolysis of legume-derived proteins) on the expression of genes involved in auxin signaling (*SlIAA2* and *SlIAA9*) in leaf tissue of tomato plants were compared with distilled water (negative control) and indole-3-acetic acid (positive control) [62]. *SlIAA9* was upregulated by four PHs and indole-3-acetic acid, while *SlIAA2* was upregulated by the commercial product and the Brassicaceae-derived PH, assuming that these PHs might elicit auxin-like activity to regulate root architecture [62]. In tomato cuttings, the foliar application of (Malvaceae and Solanaceae) plant-derived PHs positively influenced root length, partially through the interaction between auxins and gibberellins, that both accumulated in response to Solanaceae- and Malvaceae-derived PHs [63]. This effect was also reported for a PH derived from a tanning industry by-product, that modulated the expression of genes involved in the gibberellic acid (GA) metabolism (upregulation of the gene encoding gibberellin 3-beta-dioxygenase 1 and downregulation of the gene encoding gibberellin 2-oxidase) in maize seedlings [64]. GA can regulate auxin transport by adjusting PIN-Formed (PIN) proteins abundance — key auxin efflux transporters in plants [63,65,66]. Foliar application of a Solanaceae-derived PH to tomato cuttings induced an elevated level of zeatin, a cytokinin playing a possible role in adventitious root extension [63,67]. AA supply experiments also evidenced changes in phytohormone profiles. In bean (*Phaseolus vulgaris*) seedlings, both asparagine and glutamine treatments increased and decreased auxin, gibberellin, and cytokinin levels at low (1 mM) and high (5 mM) concentrations, respectively. On the contrary, the abscisic acid (ABA) level decreased and increased in response to low and high concentrations of these two AAs, respectively [68].

The PH-dependent promotion of root development would also be elicited by PH peptides. Maize seedlings treated with an animal-derived PH containing 10% (*w*/*w*) of free AAs exhibited greater lateral root length and area, than maize treated with inorganic nitrogen or a reconstituted mixture of free AAs from the same PH. Transcriptomic analyses revealed that the animal-derived PH could act by regulating a glutamate receptor involved in root growth and C/N signaling, in contrast to the reconstituted mixture of free AAs. Furthermore, *GRMZM2G055607_T01*, encoding a sulfotransferase that catalyzes the post-translational tyrosine sulfation of secreted peptides, was over-expressed in maize plants treated with the same PH, compared to AA-treated plants [64]. Tyrosylprotein sulfotransferase is essential for root development, as evidenced by the short-root phenotype of the loss-of-function mutant of this gene (*tpst-1*) due to a sharp decrease in the number of proximal meristem cells [69,70]. A differential regulation of genes encoding CLE peptides and CLE receptor kinase CLAVATA1 (CLV1) in response to PH- and AA-treated maize was reported [64]. CLAVATA3 (CLV3)/EMBRYO SURROUNDING REGION-related(CLE) family peptides act as mediators of cell-to-cell communication. Members of this family are implied in the differentiation of shoot and root meristems [71]. From a mechanistic point of view, signaling peptides bind to membrane receptors and trigger signaling cascades responsible for the fine tuning of different processes of root development, including cell expansion or the emergence of lateral roots and root hairs [71,72,73,74]. The action of PHs on root architecture seems to differ from that of isolated AAs [64]. The exogenous supply of glycine can influence root morphology by inhibiting root elongation in pak choi (*Brassica campestris* ssp. *Chinensis* L.) [75]. Isolated AAs, including leucine (L-Leu), lysine (L-Lys), tryptophan (L-Trp), and glutamate (L-Glu), repress cell division and elongation in Arabidopsis primary roots [76], and its effect could be associated with auxin- and mitogen-activated protein kinase (MAPK) signaling [76,77]. All these findings indicate that PHs can improve the root system architecture, and future investigations will be performed to decipher regulatory molecular networks and understand cross-talks between hormones and peptides in PH-controlled root physiology.

### 3.2. Shoot Biomass and Yield

Applications of PHs increased the leaf area and the yield of some fruit trees and other horticultural plants [78,79,80]. In this respect, the available literature data highlight a positive effect of the PHs’ application, isolated AAs (proline, tryptophan) or mixtures of AAs (alone or combined with micronutrients) on flowering regulation, fruiting, and fruit quality [81,82,83,84,85,86]. For instance, application of an animal-derived PH (enzymatic hydrolysates obtained from animal hemoglobin) resulted in earlier flowering, and in a significant increase in early fruit production in cold-stressed strawberry plants [87]. Better growth and development of two crops (Brinjal and chilli plants) with early flowering and an elevated yield has been reported in response to an exogenous supply of an animal-derived PH (feather hydrolysate) [88]. The production of petunias with extra-grade visual quality was also enhanced by foliar application of an animal-based PH [89]. Exogenous applications of a mixture of AAs (20%) and algal extracts (12%) (‘Primo’) on grapevine resulted in a higher number of bunches per cane and greater berry weight and size compared to the control [90]. The quality of citrus fruit was also improved by this same mixture [91]. PHs (e.g., Trainer^®^) and a biostimulant containing free AAs (e.g., CycoFlow) increased the total yield (number of fruit) and nutritional value of tomato [25,43,92], with a positive effect on pollen viability [92]. PHs can also induce secondary metabolism activity (carotenoids and polyphenols) [26].

Although the mode of action is still unknown, PHs might stimulate auxin- and gibberellin-like activities [56,59]. Gibberellin regulates floral transition and flowering [93,94] and auxin is involved in floral opening [95]. The precursor of ethylene—1-aminocyclopropane-1-carboxylate (ACC)—was also accumulated in tomato plants treated with plant-derived PH [96]. However, it was reduced in lettuce plants grown under control conditions and treated with the commercial PH Vegamin^®^ [97]. Ethylene is often associated with senescence and fruit ripening, but it could also promote flowering in some species, while inhibiting it in others [98,99]. In addition, some AAs are precursors of components responsible of aroma (alanine, isoleucine, leucine, and valine) and color (precursors of anthocyanin biosynthesis) [26]. Additional reports indicate that exogenous AA (e.g. L-arginine, L-cysteine, L-methionine, and GABA (gamma-aminobutyric acid)) application could also participate in postharvest vegetable and fruit quality maintenance [100,101]. These findings indicate that the beneficial effects of PHs are not restricted to the vegetative phase of plants but also concern their reproductive phase.

## 4. Effects of PHs on Plant Physiology

PHs, that have positive effects on crop performance [102], can improve nitrogen assimilation (N) and photosynthetic activity, two processes that highly influence shoot meristematic activity (Figure 1).

### 4.1. Photosynthetic Activity

The response of photosynthetic activity to PH and/or AA supply results from the convergence of fine-tuned changes in cell metabolism. The application of AAs induced a significant increase in total chlorophylls (chl a + b) and carotenoids in wheat leaves [103]. In broccoli, AA application increased the photosynthetic rate, stomatal conductance, the internal CO_2_ concentration, and the transpiration rate in the ‘Agassi’ cultivar prior to stress application [104]. Foliar glycine betaine (GB) application (5 mM) guaranteed an elevated photosynthetic and transpiration rate, a high intracellular CO_2i_ concentration, and high stomatal conductance (g_s_) in cotton seedlings despite exposure to salt stress [105]. Proline and L-pyroglutamic acid (a non-protein amino-acid derivative) application could also play a role in maintaining the photosynthetic rate, in turn leading to improved plant growth under abiotic stress [106,107]. A positive effect of L-tyrosine, L-lysine, L-methionine, or L-arginine application on the photosynthetic activity of tomato or *Pereskia aculeata* Mill. has also been reported [41,108]. PHs (plant- or animal-derived) have a positive effect on the photosynthetic performance of crops grown under favorable and/or unfavorable conditions including different N fertigation levels or salt stresses [43,97,109,110]. For instance, the application of alfalfa-derived PH resulted in chlorophyll production and upregulation of genes encoding components of the photosynthetic electron transfer chain (ferredoxin-2, the light-harvesting complex protein LHCA5) and Calvin cycle enzyme ribulose-1,5-bisphosphate carboxylase/oxygenase (RuBisCo) in tomato plants [111]. In line with this, an elevated sugar content has been reported in tomato plants and wheat seedlings after the application of PHs (an alfalfa-derived PH or a chicken feather-derived PH) [111,112]. Additional investigations pointed out an action of alfalfa-derived PH on the transcript accumulation of key genes of the major carbon metabolism, including phosphoenolpyruvate carboxylase, malate dehydrogenase and fumarate dehydrogenase. Increase in production of carbon skeletons could stimulate the N assimilation in plants [111,113]. All these findings indicate that PHs lead to a finely tuned regulation of photosynthesis, primary carbon metabolism, and N assimilation, which altogether contributes to high plant biomass and growth. It will be relevant to investigate the effect of these PHs on the accumulation of sugar transporters (Sugars Will Eventually Be Exported Transporter (SWEET) and phloem-located sugar transports (Sucrose transporters)), which play a key role in the translocation of photosynthates from source to sink organs [114]. The impact of PHs on plant branching is also not known yet, although it is an integral part of plant yield and agronomic performance [115].

### 4.2. Nutrient Uptake and Assimilation

#### 4.2.1. Nitrogen Acquisition and Assimilation

Aside from their positive effects on root system architecture, PHs could promote plant growth by stimulating nitrogen uptake and assimilation [59,111,113]. Organic nitrogen can also significantly contribute to increase nitrogen use efficiency (NUE) in crop production, as its assimilation into proteins has a lower carbon cost than that of inorganic nitrogen [116]. Substrate drench with the plant-derived PH Trainer^®^ enhanced the expression of the key gene *AAT1* that encodes an AA transporter involved in the transport of glutamic acid, aspartic acid, and isoleucine, in leaves and roots of tomato plants [58]. The application of gelatin capsules near cucumber seeds enhanced the expression of genes encoding AA transporters (e.g., amino acid permeases), notably AAP3 and AAP6, contributing to elevated N uptake [117]. AAP3 is involved in basic AA transport, and AAP6 efficiently transports neutral and acidic AAs [117,118,119,120]. However, the applied AA dose is generally low, so that positive effects were only attributed to increased N availability [113], highlighting that the beneficial effects of PHs to plants are multi-factorial [121]. Regarding inorganic nitrogen, the legume-derived PH Trainer^®^ downregulated the expression of genes encoding two high-affinity nitrate transporters (*NRT2.1* and *NRT2.3*), whereas an alfalfa-derived PH upregulated the nitrate transporter (NTP2, a homologous of *Arabidopsis thaliana* AtNRT1:4) gene in tomato plants [58,111]. *NRT2.1*—an inducible high-affinity nitrate transporter gene—is involved in the repression of lateral root initiation in response to nutritional cues [122], and its PH-dependent repression might explain the improvement in root development [58]. *NRT2.3* may be involved in nitrate uptake and the root-to-shoot long-distance transport in tomato [58,123], while *AtNRT1:4* is expressed in leaves and involved in leaf nitrate homeostasis within the plant [124]. In maize seedlings, the effect of a PH derived from tanned bovine hides (APR^®^) on transcript levels of genes encoding high-affinity nitrate transport systems (*NRT2* genes) seems to vary according to the growing conditions [125]. Altogether, this fits with the decrease in nitrate and ammonium influx in roots with exogenously supplied AAs [126], and the reduced nitrate uptake by soybean seedlings treated with certain AAs [127]. In line with these findings, it was suggested that glutamine downregulates the expression of *HvNRT2*, which encodes high-affinity nitrate transporters in barley [128]. Additional investigations on the relationship between PH treatment and nitrogen uptake are required to shed light on the underlying molecular mechanisms—a fundamental step in view of improving NUE under stressful environmental conditions. Several reports highlight a positive effect of PHs on nitrogen assimilation processes [53,58,59,113], and this effect is common to animal- and plant-derived PHs [53]. An AA-rich biostimulant (Aminoplant/Siapton^®^, an animal-derived PH) improved nitrate reductase activity in maize grown under moderate salt stress [129,130]. In a high nitrogen regime, legume seed-derived PH Trainer^®^ increased the expression of N assimilation-related genes (encoding nitrate reductase, nitrite reductase, and ferredoxin-dependent glutamate synthase) in tomato roots [58]. An alfalfa-derived PH stimulated the expression of genes encoding nitrate reductase, aspartate aminotransferase, glutamine-dependent asparagine synthetase, glutamine synthetase, and N-associated genes related to AA synthesis and turnover (glutamate dehydrogenase, serine decarboxylase) and protein accumulation (translation initiation factors) in tomato plants [111]. PHs Amino16^®^ and Trainer^®^ are thought to activate AA remobilization and turnover and ammonium recycling, which helps plants to mitigate physiological disorders caused by nitrogen excess [131,132]. This hypothesis is supported by the elevated production of chlorophylls and/or proteins [49,52,133] along with a high sugar content [49,111] providing carbon skeletons for protein synthesis [16].

#### 4.2.2. Macroelements: S, P, K, Mg, and Ca 

Regarding sulfur nutrition, an alfalfa-derived PH positively affected plant sulfur uptake by upregulating genes coding for sulfate transporters in tomato plants [111,134]. A biostimulant composed of a mixture of AAs (proline and tryptophan), an *A. nodosum* extract, and a lignosulfonate increased potassium uptake by almond plants, especially in K-depleted soil [135]. Investigating the impact of the foliar application of three types of PHs (native whey protein (NAP), papain- (PAH-), or pepsin- (PEH-) hydrolyzed whey protein) on N, P, and K uptake by pea (*Pisum sativum*) plants, it was demonstrated that these biostimulants enhanced the uptake of these macroelements [136]. An alfalfa-based PH increased the expression of genes coding phosphate transporter PT2 and potassium channels in tomato plants [111]. Other biostimulants seem to promote the uptake of phosphorus and/or potassium, e.g., fulvic acids (^32^P uptake) in wheat [137] and *Aphanothece sp*. in tomato plants [138]. Mg and Ca acquisition and assimilation by spinach [139] and maize plants [52] were enhanced by Trainer^®^ and a chickpea-derived PH, respectively.

#### 4.2.3. Microelements

The positive relationship between PHs and plant mineral nutrition (e.g., iron) can be at least assigned to the chelating power of some of their AAs [140]. Certain amino acids (e.g., proline) could protect plants against heavy metal toxicity and also contribute to micronutrient acquisition [10]. Fe-AA complexes such as [Fe(Arg)_2_], [Fe(His)_2_], and [Fe(Gly)_2_] improved Fe accumulation in tomato roots and shoots compared to the Fe-EDTA complex, making the Fe-AA complex a powerful alternative to provide iron in nutrient solutions [141]. In tomato plants, plant-derived AAs induced a higher Fe(II)/Total Fe ratio, probably due to their action on the activity of leaf Fe(III)-chelate reductase [37]. In the same vein, maize plants under Fe deficiency and treated with a collagen-based PH mixed with FeCl_3_ displayed a faster recovery than plants treated with FeCl_3_ alone or FeEDTA. These PH-treated plants had a higher iron concentration in their leaves and a better adjustment in the expression of Fe-related genes, including those involved in Fe acquisition in roots (*ZmTOM1* and *ZmIRT1*) [142]. Certain AAs could also increase micronutrient availability by acting as a reductant. It has been hypothesized that the change in the copper oxidation state from Cu(II) to Cu(I) induced by cysteine was responsible for elevated copper uptake and translocation within maize seedlings [143]. Under alfalfa-derived PH treatment, copper transporters were upregulated in tomato plants [111]. Some peptides—components of PHs—also had a chelating capacity that improved nutrient availability and acquisition by the root system [14].

Taken together, these results indicate that PHs affect the uptake and assimilation of certain nutrients in different ways, so that the use of AAs and PHs can be an environment-friendly strategy in low-fertility soils [144,145]. This research field requires an in-depth investigation to lay the foundation for a comprehensive strategy devoted to the improvement of mineral acquisition by plants under constrained environmental conditions.

### 4.3. Metabolomic Profile Adjustment under Adverse Growth Conditions

The application of PHs also results in metabolome reprogramming in plants grown under different constrained growth conditions. After drought imposition, (plant-derived PH) GHI_16_VHL-primed *Capsicum annuum* L. plants recovered faster probably due to the higher leaf osmolyte accumulation during drought [146]. *Xcell Boost* (a mixture of fish protein hydrolysates and Kelp extract (*Ecklonia maxima*)) treatment also increased the accumulation of osmoprotectants (proline and total soluble sugars) in spinach under drought stress [147].

In line with this, the effects of the exogenous supply of GABA or proline have been under special focus, due to their “osmoprotective” character. GABA application would induce better drought tolerance in different species: perennial ryegrass [148], bread wheat [149], and creeping bentgrass (*Agrostis stolonifera*) [150]. In white clover, the increase in endogenous GABA following exogenous supply could positively regulate the GABA shunt, proline and polyamines metabolism, and improve drought tolerance [151]. This adjustment in proline content was also found in snap bean plants under water stress [152]. Moreover, in addition to its antioxidant action [106,153], proline supply led to an increase in its own endogenous level [153] notably required for osmotic adjustment. Proline acts on plant–water relationships by increasing the leaf relative water content (RWC) and cell membrane integrity in drought-stressed onion compared to untreated plants [154]. The action of other isolated exogenously supplied AAs has been investigated in different plants under water stress, notably ornithine [155], arginine [156], or glutamate [157]. Glutamate application on water-deficient lettuce had slight effects on the nitrate and proline contents of lettuce leaves [157].

### 4.4. Defense-Related Phytohormones Responses

PHs could act on mechanisms underlying the phytohormonal response to abiotic stress [21]. Under water stress, the application of the animal-derived PH Pepton increased gibberellin (GA_1_ and GA_3_), cytokinin (trans-zeatin), and auxin (indole-3-acetic acid) contents in tomato plants. Auxin stimulation would likely be linked to the presence of two auxin AA precursors—phenylalanine and tryptophan in Pepton [158]. The application of the plant-derived PH Trainer^®^ reduced the level of cytokinins [159], an antagonistic effector to ABA-dependent plant resistance mechanisms [160]. The high accumulation of ABA has been reported in the leaves of lettuce grown under water deficit and treated with Leafamine^®^ [49], and in the leaves of grapevine exposed to a mixture of AAs and drought stress [161]. It has been shown that GABA application resulted in ABA accumulation in the leaves of apple seedlings and the elevated expression of ABA-related genes, such as genes encoding an ABA receptor (*PYL4*), ABA-responsive element-binding factor 3 (*ABF3*), and *OST1* (encoding a protein kinase involved in ABA-mediated stomatal closure) [162]. The application of the PH Delfan Plus (with an AA-based formulation) to *Arabidopsis thaliana* stimulated the ethylene signaling pathway through the upregulation of gene *ERF1A*—coding Ethylene Response Factor 1A (transcription factor) —24 h after treatment [163]. Pepton-treated tomato plants displayed higher jasmonic acid (JA) levels under water stress compared to untreated plants [158], that could contribute to the higher expression of stress responsive genes and increased production of plastochromanol-8 conferring better protection against oxidative stress [158]. Plastochromanol-8 (PC-8), an antioxidant belonging to the “tocochromanols” group, together with tocopherols and carotenoids, could contribute to the protection of photosystem II (PSII) from environmental stress-dependent damage [164]. PSII is indeed easily damaged by an array of stressors, leading to altered photosynthetic activity [165]. Along with JA, salicylic acid could confer the plant’s tolerance to water stress [166] by modulating proline biosynthesis and maintaining the cellular redox status [167]. This could be consistent with the fact that Trainer^®^-dependent tomato resistance to water stress was related to the salicylic acid-mediated regulation of ROS accumulation [159]. Under suboptimal growth conditions, an enzymatically hydrolyzed animal protein-based biostimulant (Pepton) might enhance the primary and lateral root growth of tomato plants through the stimulation of salicylic acid biosynthesis [168]. All these findings should be used as a solid foundation to identify the mode of action of PHs on the hormone metabolism and signaling pathways.

### 4.5. ROS Metabolism Adjustment

On the whole, the exogenous application of PHs and isolated amino acids improves plant antioxidant performance by stimulating the enzymatic and non-enzymatic antioxidant defense machinery of the cell [13,23,26]. This was especially revealed for plants grown under stressful conditions. Under water deficit conditions, a biostimulant containing free AAs (CycoFlow) improved the yield of tomato plants grown under water deficit conditions by maintaining the water status, pollen viability, and reducing oxidative stress [169]. This action against oxidative stress is probably due to the presence of glycine betaine and proline [170], two well-known protectors of plants against water stress [171,172,173,174]. The application of alfalfa-based PH induced antioxidant activity in tomato plants [111]. In tomato, it was associated with the upregulated expression of several gene-encoding enzymes involved in the glutathione/ascorbate detoxifying cycle as well as the higher production of phenol compounds [111]. A pig blood-derived PH also improved the antioxidant properties of lettuce through the upregulation of several genes encoding phenol-biosynthesizing enzymes [175] and antioxidant enzymes, as well as through the accumulation of non-enzymatic antioxidant systems (flavonoids, ascorbic acid, and glutathione) in water stress-exposed tomato [176]. In line with this, PH-treated tomato plants grown under water stress exhibited a high tolerance to increased ROS assigned to the coordinated action of signal compounds, radical scavengers (carotenoids and prenyl quinones), as well as a reduced biosynthesis of tetrapyrrole coproporphyrins [159]. Some biostimulants containing AA and peptides can increase the content of anthocyanins, that may serve as antioxidants [177], in grape juice or fruit [178,179].

## 5. Conclusions and Future Challenges

PHs have a significant potential to improve not only the agronomic performance of several crop species but also their resistance to stressful conditions. Many reports point out that the beneficial effect of PH application covers many processes of plant physiology throughout their lifecycle and, thus, increases yield and quality parameters. PH application improves vegetative plant growth; plant nutrition including nutrient use efficiency, nutrient uptake, and assimilation; and fruit set and size in many crops. Interestingly, PHs would decrease nitrate accumulation in leafy vegetables. Under adverse conditions, PHs confer greater resilience to plants by protecting the activity of the photosynthetic machinery and stimulating multiple protective processes related to antioxidant activity and osmotic adjustment. However, the mechanisms driving the beneficial effects of PHs on plants are not completely understood, although the recent use of multi-omics strategies could be of great help to shed light on their modes of action. It is more likely that the action of PHs relies on interactive regulatory networks tuned by local and systemic processes. Future research will need to provide a complete picture of the mode of action of PHs (Figure 2). One challenging point is to identify the earliest mechanisms, perception and transduction pathway, behind each PH or family of PHs. This step is paramount to piece together their mode of action in a specific way. For example, the majority of PHs enhance the antioxidant machinery of plants, but the question whether they share the same triggering processes—perception, transduction, and transcription factors—or act through a specific pathway is still open. This type of research is also fundamental to identify robust physiological/molecular markers related to the efficacy of PHs and assess the physiological sensitivity of plants to the application of these products. In addition, PHs act directly as hormone-like entities and indirectly as stimulants of plant microbiomes, which may strongly contribute to the benefits derived from these products. It will be very relevant to thoroughly understand this synergy in order to implement a strategy based on the combination of PHs with specific microbial taxa characterized for their potential benefit on plant nutrition and assimilation and/or resilience to adverse conditions. All these future challenges should involve a cross-disciplinary strategy at the plant level and appropriate infrastructures capable of mimicking climate change conditions to test the relevance of PHs in the plant response to multiple stresses.

## Figures and Tables

**Figure 1 ijms-24-09714-f001:**
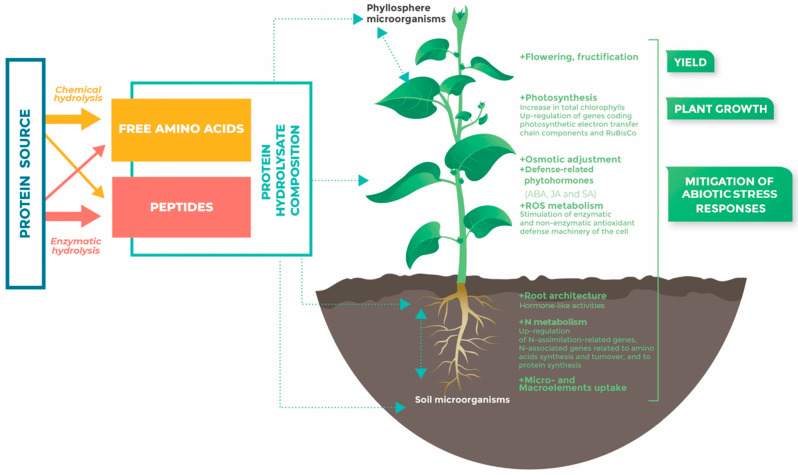
PHs’ composition and PHs’ effects on plant development and response to abiotic stress. The light blue box describes the composition of PHs (free amino acids and peptides content), depending on the hydrolysis method (enzymatic hydrolysis or chemical hydrolysis); the orange and red arrows correspond to chemical and enzymatic hydrolysis, respectively. Light blue dotted arrows indicate that PHs can be applied at the root or leaf level, and can act directly on the plant and/or indirectly through interactions with microorganisms. The green rectangles indicate the main plant responses to PHs, in terms of yield, growth, and abiotic stress mitigation. These responses were observed in the root and/or aerial parts.

**Figure 2 ijms-24-09714-f002:**
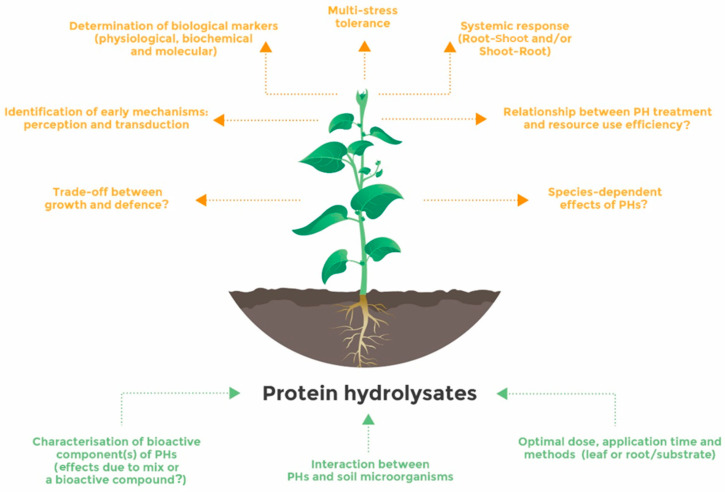
Some future research directions for the optimal and effective use of protein hydrolysates. Green arrows indicate parameters to be achieved to improve the use efficiency of PHs and brown arrows indicate future topics to be investigated to decipher the mode of action of PHs.

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
