# Peer review of "Biostimulant Properties of Protein Hydrolysates: Recent Advances and Future Challenges"

_ijms, 2023, doi:10.3390/ijms24119714_

Round 1

Reviewer 1 Report

This is a very well- written article about the class of biostimulants that are protein hydrolysates. While they have been around for years, pinpointing their effects and modes of action has been difficult. This article aims to gather together and summarize what is known about these biosimulants and propose directions for further study.

I tried, but have not thought of anything to add to this article to make it stronger, but there are a few very minor comments and a minor typographical error that is evident, namely, line 66 should start with PHs, and not Hs.

Line 48, in ‘Moreover PHs would act…’ Do you mean instead to say that they could (or may) act?

Line 164, what is a CLE peptide?

Line 386, instead of ‘plants’, suggest ‘the plant’s’ tolerance…

(References will need a bit of technical editing between Title Case and Sentence case titles)

Author Response

Dear Editor-in-Chief

We are pleased to submit the revised version of the review entitled "Biostimulant properties of protein hydrolysates: recent advances and future challenges" for publication in the International Journal of Molecular Sciences (IJMS). 
In this revised version, the recommendations and suggestions of the reviewers have been thoroughly taken into account.  
We hope that this revised version of the review will be considered suitable for publication in IJMS and look forward to your comments.

Yours sincerely,
Pr. Soulaiman Sakr

Reviewer 2 Report

This manuscript is a review article in which the authors describe in detail what biostimulants are and their effects on plant morphological characteristics and physiological processes in plants. They focus on protein hydrolysates (PH). They describe in detail the origin of protein hydrolysates and the process of their production. Hydrolysates can be mixtures of amino acids, oligo- or polypeptides formed by gradual hydrolysis of proteins from various sources. Approximately 20 million tons of milk proteins, proteinaceous seafood waste, and livestock byproducts are released into the environment each year. The use of these proteins for the production of biostimulants is part of the circular economy concept that contributes to greater sustainability in agriculture.
In the following, the authors describe the structure of protein hydrolysates and the amino acids that make them up, depending on their origin. They describe how pH affects plant growth, root system growth, above-ground plant growth and biomass, and yield. They describe the effects of PH on the physiological parameters of plants, such as photosynthetic activity, the uptake of nutrients, especially nitrogen, other macroelements such as sulfur, phosphorus, potassium, magnesium and calcium, microelements; on the regulation of metabolites under adverse stress conditions; show the effects of PH on the action of phytohormones for defense, the regulation of metabolism of ROS.
This type of research is also essential to identify robust physiological/molecular markers for the efficacy of PHs and to evaluate the physiological sensitivity of plants to the application of these products. In addition, PHs act directly as hormone-like entities and indirectly as stimulants of the plant microbiome, which may contribute significantly to the benefits of these products. It will be very important to thoroughly understand this synergy in order to implement a strategy based on combining PHs with specific microbial taxa characterized for their potential benefits to plant nutrition and as simulation and/or resistance to adverse conditions. All of these future challenges should include a cross-disciplinary plant-level strategy and appropriate infrastructures capable of simulating climate change conditions to test the importance of PHs in plant response to various stressors.

Author Response

Reviewer 2

Comments and Suggestions for Authors

This manuscript is a review article in which the authors describe in detail what biostimulants are and their effects on plant morphological characteristics and physiological processes in plants. They focus on protein hydrolysates (PH). They describe in detail the origin of protein hydrolysates and the process of their production. Hydrolysates can be mixtures of amino acids, oligo- or polypeptides formed by gradual hydrolysis of proteins from various sources. Approximately 20 million tons of milk proteins, proteinaceous seafood waste, and livestock byproducts are released into the environment each year. The use of these proteins for the production of biostimulants is part of the circular economy concept that contributes to greater sustainability in agriculture.
In the following, the authors describe the structure of protein hydrolysates and the amino acids that make them up, depending on their origin. They describe how pH affects plant growth, root system growth, above-ground plant growth and biomass, and yield. They describe the effects of PH on the physiological parameters of plants, such as photosynthetic activity, the uptake of nutrients, especially nitrogen, other macroelements such as sulfur, phosphorus, potassium, magnesium and calcium, microelements; on the regulation of metabolites under adverse stress conditions; show the effects of PH on the action of phytohormones for defense, the regulation of metabolism of ROS.
This type of research is also essential to identify robust physiological/molecular markers for the efficacy of PHs and to evaluate the physiological sensitivity of plants to the application of these products. In addition, PHs act directly as hormone-like entities and indirectly as stimulants of the plant microbiome, which may contribute significantly to the benefits of these products. It will be very important to thoroughly understand this synergy in order to implement a strategy based on combining PHs with specific microbial taxa characterized for their potential benefits to plant nutrition and as simulation and/or resistance to adverse conditions.

 All of these future challenges should include a cross-disciplinary plant-level strategy and appropriate infrastructures capable of simulating climate change conditions to test the importance of PHs in plant response to various stressors.

Reply: We appreciate and thank you for your relevant comment about this review.

Reviewer 3 Report

ijms-2389753

Comments for Authors

Authors reviewed in detail the effects of protein hydrolysates that can be used in the agriculture. The “recent advances” in the title were detailed in the text but there is a little hint to future tasks. I propose revising Fig.1, summarizing future tasks in a separate chapter, checking the text and improving the English style.

Detailed comments

66 line: there is a typing error. The correct word is “PHs”

108-109 lines: I don’t think these compounds are closely related to protein sources see the purple section on Fig 1. It is true, that glycoproteins comprise carbohydrates and protein and they act mainly in immune processes of animal and human cells. The protein-phenol interaction is not typical in plants as bio-stimulants they are more stable in food processing.

Based on this, I propose to reconsider and revise the Fig.1.

149-150 lines: The sentence is unclear. I should be rephrased. What does the opposite trend for ABA increased or decreased?

206 line: Full name of GABA should be written out for the first time. Is it gamma-aminobutyric acid?

English style correction required.

Author Response

Comments for Authors

Authors reviewed in detail the effects of protein hydrolysates that can be used in the agriculture. The “recent advances” in the title were detailed in the text but there is a little hint to future tasks. I propose revising Fig.1, summarizing future tasks in a separate chapter, checking the text and improving the English style.

Answer: We thank you for this relevant suggestion, and we decide to present future tasks on a new figure (figure 2).

Detailed comments

* 66 line: there is a typing error. The correct word is “PHs”

Answer: We made the change.

108-109 lines: I don’t think these compounds are closely related to protein sources see the purple section on Fig 1. It is true, that glycoproteins comprise carbohydrates and protein and they act mainly in immune processes of animal and human cells. The protein-phenol interaction is not typical in plants as bio-stimulants they are more stable in food processing. Based on this, I propose to reconsider and revise the Fig.1.

Answer: We thank you for this suggestion, and revised figure 1.

*149-150 lines: The sentence is unclear. I should be rephrased. What does the opposite trend for ABA increased or decreased?

Answer: We made the change : "In bean (Phaseolus vulgaris) seedlings, asparagine and glutamine treatments increased and decreased auxin, gibberellin and cytokinin levels at low (1 mM) and high (5 mM) concentrations, respectively. On the contrary, abscissic acid level decreased and increased in response to low and high concentrations of these two AA, respectively." Line 144-148

*206 line: Full name of GABA should be written out for the first time. Is it gamma-aminobutyric acid?

Answer: We added its full name. Line 207

*Comments on the Quality of English Language

English style correction required.

Answer: Has been done.